# Galectins in Equine Placental Disease

**DOI:** 10.3390/vetsci10030218

**Published:** 2023-03-13

**Authors:** Carleigh E. Fedorka, Hossam El-Sheikh Ali, Mats H. T. Troedsson

**Affiliations:** 1Department of Veterinary Science, College of Agriculture, Food and Environment, University of Kentucky, Lexington, KY 40546, USA; 2Department of Theriogenology, College of Veterinary Medicine, Mansoura University, Dakahlia 35516, Egypt

**Keywords:** placentitis, galectin, equine, chorioallantois

## Abstract

**Simple Summary:**

Galectins are a cohort of proteins that function throughout the body. These proteins are involved in reproduction, including pregnancy. Infection of the placenta can lead to abortion, but it is unknown if this coincides with alterations in galectin expression. The goal of this study was to evaluate the profile of galectins during placental infection in the horse. To do so, the placenta from mares experiencing different types of placental infection was collected, and the quantity of various galectins was assessed. The profile of a number of galectins was found to shift following placental infection, and this was dependent on the galectin assessed as well as the specific type of placental infection. This information allows us to further our understanding of these diseases, as well as warrants attention as any alterations of galectin profile may reveal potential indicators of disease.

**Abstract:**

Galectins are proteins that bind to glycans in targeted cells and function in cell-to-cell signaling throughout the body. Galectins have been found to be involved in various reproductive processes, including placental dysfunction, but this has not been investigated in the horse. Therefore, the objective of this study was to assess alterations in galectin expression of the abnormal placenta in pregnant mares. Next-generation RNA sequencing was performed on the postpartum chorioallantois of two placental pathologies following clinical cases of ascending placentitis (n = 7) and focal mucoid placentitis (n = 4), while chorioallantois from healthy postpartum pregnancies (n = 8; 4 control samples per disease group) served as the control. When evaluating ascending placentitis, both galectin-1 (*p* < 0.001) and galectin-3BP (*p* = 0.05) increased in the postpartum chorioallantois associated with disease, while galectin-8 (*p* < 0.0001) and galectin-12 (*p* < 0.01) decreased in the diseased chorioallantois in comparison with those in the control. In mares with focal mucoid placentitis, numerous galectins increased in the diseased chorioallantois, and this included galectin-1 (*p* < 0.01), galectin-3BP (*p* = 0.03), galectin-9 (*p* = 0.02), and galectin-12 (*p* = 0.04), in addition to a trend toward increases in galectin-3 (*p* = 0.08) and galectin-13 (*p* = 0.09). In contrast, galectin-8 expression decreased (*p* = 0.04) in the diseased chorioallantois in comparison with that of the controls. In conclusion, galectins alter in abnormal placentae with variations observed among two forms of placental pathologies. These cytokine-like proteins may further our understanding of placental pathophysiology and warrant attention as potential markers of placental inflammation and dysfunction in the horse.

## 1. Introduction

Galectins are a group of soluble proteins with a conserved carbohydrate recognition domain (CRD) that bind to glycans in targeted cells with high specificity. These proteins assist in cell-to-cell signaling in a cytokine-like manner and can stimulate cell adhesion, apoptosis, and regulation of both the innate and adaptive immune responses [1,2]. In women, select galectins have been found to be involved in numerous aspects of reproductive physiology, including implantation; the immunotolerance of the semiallogeneic fetus; placentation; and endocrine–immune interactions [3,4,5,6]. It is thought that galectins are involved in the successful establishment and maintenance of pregnancy, and they are routinely used as biomarkers to predict pregnancy-related complications in humans, including pre-eclampsia [7,8,9], chorioamnionitis [10,11], preterm premature rupture of membranes (PPROM) [12], and unexplained miscarriage [13,14,15,16]. While the profile of various galectins was assessed during normal pregnancy in the horse [17], no research has been published on equine galectins in mares with placental disease.

A leading cause of late-term abortion in the horse is placental inflammation/infection, deemed placentitis [18,19]. Various types of placentitis have been described in the literature, including ascending, hematogenous (EHV-1/leptospirosis), and focal mucoid (nocardioform), and yet the pathophysiology of these diseases is not fully understood [20,21]. While ascending placentitis is thought to be initiated by the ascending migration of pathogens through the vaginal canal and cervix to localize on the cervical pole of the placenta, and pathogens causing hematogenous placentitis are spread through circulation, to date, no etiology for focal mucoid placentitis has been confirmed [22]. Regardless, each disease has the potential to result in preterm delivery, abortion, or a dysmature neonate. Due to this, considerable emotional and economic consequences are noted due to these diseases. Currently, gold-standard diagnostics for these diseases include premature mammary gland development, vulvar discharge, and the use of ultrasonography to monitor the thickness of the placenta, alongside noticing if any separation occurs between the endometrium and chorioallantois [23,24]. As these clinical changes occur during very acute phases of disease, these alterations provide minimal opportunity for therapeutic intervention. Research focuses have shifted toward the detection of biomarkers within noninvasive sampling procedures, of which galectins are being utilized as in other species.

Therefore, evidence is needed to assess the involvement of galectins in placental pathologies in the horse. We hypothesized that the chorioallantois expression of specific galectins alters following placental infection and that these galectins may be pathology-specific. The main objective of this study was to assess alterations in the expression of select galectins in the postpartum chorioallantois of diseased animals. With increased knowledge of deviations from the normal chorioallantois, inferences may be made into potential markers for fetal viability and pregnancy complications or be utilized to predict when therapeutic intervention is required.

## 2. Materials and Methods

### 2.1. Animal Use and Tissue Collection

All procedures involving animals were approved by and conducted in accordance with the Institutional Animal Care and Use Committee of the University of Kentucky (Protocols #2014–1215 and 2014–1341). All horses (*Equus caballus*) used in this study were thoroughbred mares of mixed parity ranging from 450 to 600 kg and from 4 to 20 years of age residing in central Kentucky. Postpartum chorioallantois (CA) was collected from ascending placentitis (n = 7) and focal mucoid placentitis cases (n = 4) in clinical cases. Disease confirmation was performed by gross pathological and histological evaluation at a local diagnostic laboratory, as previously described by El-Sheikh et al. (2021) [21]. Control samples were collected from mares on matching farms identified to have a normal pregnancy (n = 8; 4 controls per disease group). Following parturition, the chorioallantois from normal mares was also assessed via gross pathology and histologically to ensure that the placenta was free of infection/inflammation by the same diagnostic laboratory. Each dataset had its own control samples, with n = 4 normal chorioallantois in each comparison group. For the diseased samples, tissue was obtained from the site of the lesion. For controls involved in the ascending placentitis comparison, tissue was obtained from normal chorioallantois at the cervical pole. For controls involved in the focal mucoid placentitis comparison, tissue was obtained from the normal chorioallantois at the ventral body. All tissues were stored in RNAlater (Thermo Fisher Scientific, Waltham, MA, USA).

### 2.2. RNA Isolation

RNA was isolated from postpartum chorioallantois using an RNeasy Mini Kit (Qiagen, Gaithersburg, MD, USA), per the manufacturer’s instructions and as previously described by El-Sheikh Ali et al. [25]. The purity, integrity, and concentration of RNA were analyzed by a Bioanalyzer^®^ (Agilent, Santa Clara, CA, USA). All samples had a 230/260 ratio >1.8, a 260/280 ratio >2.0, and an RNA integrity number >8.0. Following this, library preparation of RNA was performed using a TruSeq Stranded mRNA Sample Prep Kit (NEB, San Diego, CA, USA, as per the manufacturer’s instructions. To confirm purity, all reads were quantified with qPCR.

### 2.3. RNA Sequencing

Next-generation RNA-seq was carried out on postpartum chorioallantois. In brief, sequencing libraries were generated using a NEBNext^®^ Ultra™ RNA Library Prep Kit for Illumina^®^ (NEB, San Diego, CA, USA) following the manufacturer’s recommendations, as previously described by El-Sheikh Ali et al. [25]. In brief, library preparations were sequenced on a NovaSeq 6000, and 125 bp/150 bp paired-end reads were generated. Following this, reads were trimmed for quality and then mapped to EquCab3.0 using hisat2 v2.1 [26]. Expression values (fragments per kilobase measured (FPKM)) of mapped reads were quantified using FeatureCounts for v1.5.0 [27] with the NCBI annotation (GCF_002863925.1). Expressions of galectin-1, -2, -3, -3BP, -4, -5, -6, -7, -8, -9, -9B, -10, -11, -12, -13, -14, -15, -L, and GRIFIN were assessed. For ascending placentitis, the RNA-seq data were deposited in the Gene Expression Omnibus (GEO; GSE166617) repository [28]. For the focal mucoid placentitis, the RNAseq data were also deposited in the GEO (GSE154637) repository.

### 2.4. Statistics

To compare galectin expression levels between diseased and control placenta, statistical analyses were performed utilizing SAS 9.4 (SAS Institute, version 12.1.0). Comparisons of FPKM were performed using an unequal variances t-test, with post hoc analysis by Tukey’s HSD to evaluate the effect of disease. Descriptive statistics are shown as mean ± SE unless otherwise stated. Significance was set to *p ≤* 0.05.

## 3. Results

### 3.1. Galectin Expression in Ascending Placentitis

Ascending placentitis was determined based on the identification of gross and histopathological abnormalities located at the predilection site for the disease within the cervical star region of the chorioallantois alongside positive culture/PCR for *Streptococcus equi ssp. zooepidemicus* within the lesion, as previously described by Hong et al. (1993) [29]. To evaluate the expression of the galectins evaluated in the postpartum chorioallantois of ascending placentitis, comparisons were made between diseased tissue obtained from the cervical star region of the chorioallantois and the cervical star region of healthy chorioallantois that was determined to be free of placental disease (Table 1/Figure 1). The expression of galectin-1 was dramatically increased in mares with ascending placentitis in comparison with that of the control (*p* < 0.001). In diseased placenta, galectin-1 experienced a six-fold increase in comparison with that of the controls. A significant increase in the expression of galectin-3BP was also noted in the ascending placentitis group in comparison with that in the controls (*p* = 0.05). This is in contrast to galectin-8 (*p* < 0.0001) and galectin-12 (*p* < 0.01), where a significant decrease in galectin expression was noted in the diseased placenta compared with that of the controls. No significant alteration was observed for the expressions of galectin-3 (*p* = 0.26), galectin-4 (*p* = 0.24), or galectin-13 (*p* = 0.13) in the ascending placentitis group compared with those of the controls.

### 3.2. Galectin Expression in Focal Mucoid Placentitis

Focal mucoid placentitis was determined based on the gross and histopathological abnormalities at the predilection site of the disease within the body of the diseased placenta, in addition to a positive culture/PCR for *Amycolatopsis Lexingtonian,* as previously described [29]. To assess the effect of focal mucoid placentitis on the expression of various galectins, comparisons were made between the tissue isolated from the lesion itself and the matched chorioallantois of placenta identified as having no disease (Table 2/Figure 2). The majority of galectins assessed significantly increased within the lesion. This included galectin-1 (*p* < 0.01), galectin-3BP (*p* = 0.03), galectin-9 (*p* = 0.02), and galectin-12 (*p* = 0.04). Additionally, a trend toward a significant increase was observed for galectin-3 (*p* = 0.08) and galectin-13 (0.09). Similar to ascending placentitis, a significant decrease in the expression of galectin-8 (*p* = 0.04) was noted in the focal mucoid-infected placenta samples compared with that of the controls. The expression levels of galectin-4 were undetectable in the majority of samples; therefore, conclusions could not be made.

## 4. Discussion

Galectins are didactic upstream regulators of a variety of perturbations in the feto–maternal interface during pregnancy. This includes stress, infection, malnourishment, and endocrine atrophy, all of which are detrimental to fetal growth and development. In this study, we evaluated the profile of various galectins following two common types of equine placental infection: ascending placentitis and focal mucoid placentitis. To the best of our knowledge, the galectin profile following equine placentitis has not been previously described.

Like cytokines, galectins can have proinflammatory, pleiotropic, or anti-inflammatory effects, dependent on the tissue and context of inflammation; therefore, galectins are heavily involved in the response and resolution to infectious disease. Therefore, this study investigated the expression of various galectins within the diseased placenta following both ascending and focal mucoid placentitis. Alterations were noted in the expressions of galectin-1, -3, -3BP, -9, -12, and -13; the majority of galectins increased regardless of infection type. The expression of galectin-1 was found to increase following both ascending and focal mucoid placentitis, where levels increased six-fold in the diseased postpartum placenta. This galectin was found at its highest expression or secretion during the recovery phase of inflammation [30]. As both types of infected placenta were carried to term, it could be assumed that the acute response was accomplished, and the postpartum tissue was more exemplary of a chronic inflammation. Similarly, the expression of galectin-3BP increased following both focal mucoid and ascending placentitis. It was shown that both galectin-3 and galectin-3BP are involved in the chemotaxis of leukocytes into sites of inflammation [31]. This increase in galectin-3BP may therefore be associated with the massive influx of immune cells to the site of inflammation noted in both types of placentitis [24,32].

When assessing the expression levels of galectin-3, -9, and -13, an increase was noted only in the focal mucoid-affected placenta, while the placenta associated with ascending placentitis experienced no alterations. Galectin-3 was found to bind to both pathogenic and commensal microorganisms, potentially altering the pathogenesis of microbial infections [33]. This is interesting as the pathogenicity of focal mucoid actinomycetes has not been determined [22], thereby indicating that galectin-3 may be activated by the commensal activity of the various microorganisms associated with focal mucoid placentitis. Additionally, a similar increase following focal mucoid placentitis was noted for galectin-9 expression. Upregulation of both galectin-9 and its counterpart TIM-3 has been found in pregnancy-related disorders, although it should be noted that galectin-9 decreased prior to spontaneous abortion, and its regulation may depend on the nature of the disease and the associated pregnancy outcome [9,13,34,35,36]. With regard to galectin-13, a trend toward an increase in expression was observed following focal mucoid placentitis. This protein is considered a placental alarmin due to the upregulation of its expression/secretion from the trophoblast during the clinical onset of various pregnancy-related complications, including preeclampsia, IUGR, and preterm labor [37]. It is notable that the expression of these specific galectins increased following infection associated with a focal mucoid actinomycete and not streptococcus zooepidemicus, but the explanation for this could not be determined under the confines of this study.

Unlike the other galectins evaluated, galectin-8 expression decreased in both the ascending placentitis and focal mucoid placentitis groups. In humans, galectin-8 is used as a biomarker of pregnancy health due to its involvement in T-cell proliferation, and it has been utilized for a variety of pregnancy-related complications [38]. Interestingly, increasing amounts of galectin-3 following experimentally induced streptococcus infection were found to inhibit galectin-8 production [39], and this may explain the decrease in galectin-8 expression following placental infection associated with increasing galectin-3 expression, which was noted in the present study. Galectin-12 altered following placental infection in a manner that validates the previous hypothesis that the varying pathophysiology of focal mucoid placentitis is vastly different than that of ascending placentitis, as galectin-12 expression decreased following ascending placentitis, while an upregulation of galectin-12 was noted in the chorioallantois of focal mucoid placentitis. Little is known regarding the involvement of galectin-12 in infectious disease, although it has been associated with inflammation [40]. To the best of our knowledge, no reports have associated galectin-12 with placental inflammation in any species, including equines.

This study is limited in that a small number of mares were investigated for interpretation into the two diseases. This allows for inconsistency in age, parity, nutrition, and other variables, which may have confounded these datasets. Because of this, principal component analysis (PCA) was run for each dataset, indicating that the disease and control groups were appropriately clustered. Therefore, extrapolations into the effect of disease should be considered conclusive within this study. Additionally, this study utilized the expression of RNA within the chorioallantois, and this was not validated by protein localization or production into circulation. It is acknowledged that the expression of proteins is not the production of protein, and further research would benefit an understanding of this topic. Future research is necessary to determine if any of the galectins that altered in expression within the chorioallantois following disease experience comparable shifts in protein profile within circulation that could potentially be used as noninvasive sampling biomarkers for these two diseases.

## 5. Conclusions

In conclusion, the profile of galectins at the maternal–fetal interface is dynamic in that it fluctuates following disease; yet, this alteration is dependent on the specific galectin in question. In other species evaluated, it is thought that galectins in the chorioallantois may regulate tissue development, promote antimicrobial effects, and participate in the regulation of inflammation, all of which may participate in the response to bacterial placental infection. Various galectins increased in expression following both ascending and focal mucoid placentitis, while galectin-8 expression decreased in the diseased postpartum placenta following both diseases. These alterations in the expression of galectins following placental infection warrant attention as potential markers for placental inflammation and dysfunction, as well as future studies to better understand the underlying pathophysiology of these common causes of premature delivery and abortions in mares.

## Figures and Tables

**Figure 1 vetsci-10-00218-f001:**
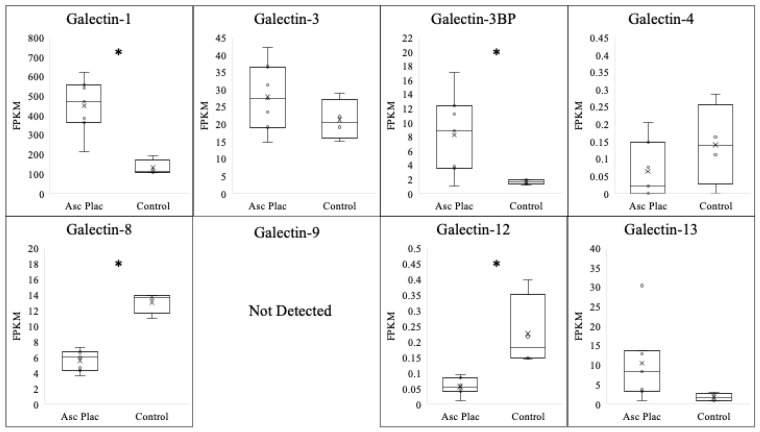
Expression profile of various galectins following ascending placentitis. Expressions of galectin-1 and galectin-3BP increased (*p* < 0.05) following ascending placentitis in postpartum chorioallantois, while expressions of galectin-3, galectin-4, and galectin-13 were unchanged. In contrast, expressions of galectin-8 and galectin-12 significantly decreased in the ascending placentitis group compared with those of controls. Expression of galectin-9 was not above the FPKM threshold for detection. * *p* < 0.05.

**Figure 2 vetsci-10-00218-f002:**
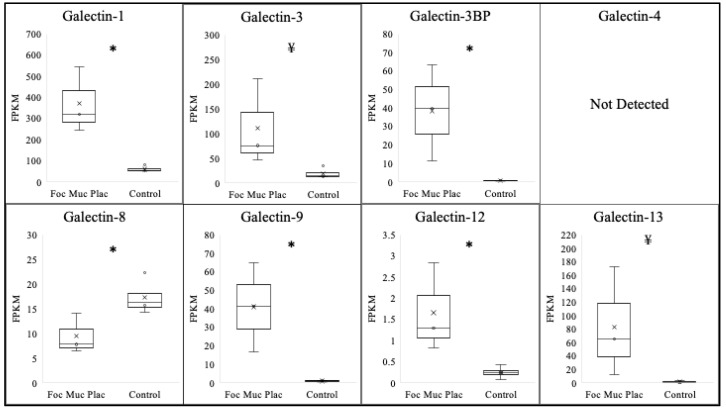
Expression profile of various galectins following focal mucoid placentitis. Expressions of galectin-1, galectin-3BP, galectin-9, and galectin-12 increased (*p* < 0.05) following focal mucoid placentitis in postpartum chorioallantois, while expressions of galectin-3 and galectin-13 experienced a trend toward an increase (*p* < 0.10). Expression of galectin-8 decreased following focal mucoid placentitis. Expression of galectin-4 was not above the FPKM threshold for detection. * *p* < 0.05. ¥ < 0.10.

**Table 1 vetsci-10-00218-t001:** Expression of various galectins following ascending placentitis. FPKM = fragments per kilobase measured. * *p* < 0.05, ND = no detectable expression.

Galectin	Normal Postpartum Chorioallantois (FPKM)	Ascending Placentitis Postpartum Chorioallantois (FPKM)	log2
**Galectin-1**	**52.3 ± 19.9**	**451.8 ± 52.3 ***	**3.10**
Galectin-3	21.4 ± 2.9	27.9 ± 3.7	N/A
**Galectin-3BP**	**1.6 ± 0.2**	**8.3 ± 2.2 ***	**2.38**
Galectin-4	0.1 ± 0.1	0.1 ± 0.1	N/A
**Galectin-8**	**13.1 ± 0.7**	**5.6 ± 0.5 ***	**−1.20**
Galectin-9	ND	ND	N/A
**Galectin-12**	**0.2 ± 0.06**	**0.1 ± 0.01 ***	**−1**
Galectin-13	1.7 ± 0.5	10.5 ± 3.8	N/A

**Table 2 vetsci-10-00218-t002:** Expression of various galectins following focal mucoid placentitis. FPKM = fragments per kilobase measured. * *p* < 0.05, ¥ < 0.10, ND = no detectable expression.

Galectin	Normal Postpartum Chorioallantois (FPKM)	Focal Mucoid Placentitis Postpartum Chorioallantois (FPKM)	log2
**Galectin-1**	**61.3 ± 6.5**	**370.6 ± 90.3 ***	2.61
**Galectin-3**	**19.0 ± 5.5**	**110.0 ± 50.9 ** **¥**	2.51
**Galectin-3BP**	**0.7 ± 0.1**	**38.3 ± 15.0 ***	5.77
Galectin-4	ND	ND	N/A
**Galectin-8**	**17.3 ± 1.7 ***	**9.5 ± 2.3**	−0.85
**Galectin-9**	**1.0** **± 0.2**	**41.1** **± 12.0 ***	5.35
**Galectin-12**	**0.2** **± 0.07**	**1.7** **± 0.6 ***	3.09
**Galectin-13**	**1.3** **± 0.6**	**83.1** **± 47.1** **¥**	6.00

## Data Availability

The data presented in this study are available in NCBI Sequence Read Archive via the Gene Expression Omnibus (GEO), accession numbers GSE166617 and GSE154637.

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
