# Peer review of "Galectins in Equine Placental Disease"

_vetsci, 2023, doi:10.3390/vetsci10030218_

Round 1

Reviewer 1 Report

General comments

An interesting paper on galectins in mares with placentitis! The research topic is new and potentially useful, and it will be interesting to see what role these galectins play in maternal recognition of pregnancy and perhaps endometritis, as well. The methodology seems appropriate. The paper is very nicely and professionally written, it is clear and easy to read. I would recommend publication of this paper, and I only have a few minor comments, the main one concerning the somewhat vague description of the control samples. They are described a bit more in the Results-section, but this part in the M&M would need some clarification (were the control samples from a different site in the same animal, or from a different animal, and were different animals somehow matched etc), and also check that there is no unnecessary repetition (in M&M vs. Results).

Detailed comments:

- line 71: remove one zero from 4500

- section 2.1.: the description of control samples is a bit unclear; what samples served as controls?

- lines 114-115: Please check this sentence for clarity; it now seems like there was a 6-fold increase in galectin-1 expression in the controls, which is probably not the case. If I understood correctly, you mean a 8.6-fold increase in diseased compared to controls?

- Table 1: please include an explanation of the abbreviation FPKM in the table legends. There is also some formatting issue in the control sample row for galectin-4 (different font?)

- lines 182 and 198 (and perhaps elsewhere?): bacterial species with capital letter and in italics

Author Response

Thank you for taking the time out of your busy schedules to review this manscript.  We have responded to each individual comment, and our responses can be seen in bold.

Reviewer 1:

An interesting paper on galectins in mares with placentitis! The research topic is new and potentially useful, and it will be interesting to see what role these galectins play in maternal recognition of pregnancy and perhaps endometritis, as well. The methodology seems appropriate. The paper is very nicely and professionally written, it is clear and easy to read. I would recommend publication of this paper, and I only have a few minor comments, the main one concerning the somewhat vague description of the control samples. They are described a bit more in the Results-section, but this part in the M&M would need some clarification (were the control samples from a different site in the same animal, or from a different animal, and were different animals somehow matched etc), and also check that there is no unnecessary repetition (in M&M vs. Results).

Detailed comments:

- line 71: remove one zero from 4500

Amended.

- section 2.1.: the description of control samples is a bit unclear; what samples served as controls?

Clarified.

- lines 114-115: Please check this sentence for clarity; it now seems like there was a 6-fold increase in galectin-1 expression in the controls, which is probably not the case. If I understood correctly, you mean a 8.6-fold increase in diseased compared to controls?

Clarified.

- Table 1: please include an explanation of the abbreviation FPKM in the table legends. There is also some formatting issue in the control sample row for galectin-4 (different font?)

Amended

- lines 182 and 198 (and perhaps elsewhere?): bacterial species with capital letter and in italics

Amended.

Reviewer 2 Report

Thank you for the opportunity to review the manuscript entitled "Galectinology of equine placental disease". The mauscript examines transcripts of the galectin family in equine placenta samples derived from clinical cases of mucoid and ascending placentitis. There are major concerns with the current publication, namely that it borrows heavily from the authors previous publications, including "Galectinology of Equine Pregnancy." Animals 13 (2023): 129. So similar are the manuscripts that the first sentences of the introduction from each are almost identical. While I acknowlege that the authors are pursuing a theme, they have somewhat undermined their own efforts.

In addition to the afore mentioned concerns, additional concerns are held for the experimental design. Specific comments are listed below.

Title: While this title matches the authors previous title, it is not entirely reflectitve of the associated data. Additionally 'galectinology' gives the impression that this manuscript gives an overview of galectins in placental disease, rather than reporting on specific galectin transcripts. Please consider reformating to include a more accurate title.

Abstract:

-The abstract indicates three groups (ascending, norcardiaform and normal). However, the methods and results appear to compare normal and diseased regions of the same placenta.Please clarify throughout.

- Please add P-values to the abstract for signficant findings.

Introduction:

- Please address concerns listed above with regard to self citation and heavy borrowing from your previous submission.

Methods:

Line 71: Extra zero in 450(0)

Line 71: With such small numbers it is probably worthwhile to provide more information about the individual horses parity and age. Both of these play an important part in placental development and should not be brushed over.

Lines 72-76: Please edit to improve readability. From abstract APL=7, NPL=4 and control=4. This is hard to follow in the methods as they are currently presented. This may also be improved by expanding your methods somewhat, rather than relying on the self citations.

Results:

- Please show PCA plots for all comparisons in both placentitis groups. Again it is unclear if the comparison of tissue is affected vs unaffected horse, or affected vs unaffected region. If it is the former then this study is grossly underpowered and the number of controls should be at least 2:1 to your cases to have robust comparisons. If it is the later then PCA by individual will help to demonstrate if disease truely drives difference in expression rather than the other variables (age, parity, sample location). 

-The  NPL gorup is really too small to perform statistical analysis and draw conclusions from. Consider including descriptive statistics only for this sextion.  Again please show a PCA even though it is limited by the sample size it is important to show.

- It would be more typical to show log2 fold changes for transcripts.

Discussion:

Although interesting to read and well written, the discussion fails to address any limitations of the current study. The most glaring of these limitations is that this set of transcripts is repeated data from the original RNA-seq analysis (Ali et al Vet Rec 2021). The galectin transcipts for NPL have already been reported and are contained in the supplementary data of that publications. This current publications offers no new analysis of these data and simply repeats that which has already been reported. I acknowledge that this manuscript is highlighting galectins as a new avenue of study, however, more independent work or analysis should have been done (qPCR, protein work etc) to advance the field.

The other major limitation is that of power. This study is very underpowered for the type of investigation. In particular there are far to few controls and this should be improved bfore resubmission.

Author Response

Thank you for taking the time out of your busy schedules to review this manuscript.  We have responded to each individual comment, and our responses can be seen in bold.

Thank you for the opportunity to review the manuscript entitled "Galectinology of equine placental disease". The mauscript examines transcripts of the galectin family in equine placenta samples derived from clinical cases of mucoid and ascending placentitis. There are major concerns with the current publication, namely that it borrows heavily from the authors previous publications, including "Galectinology of Equine Pregnancy." Animals 13 (2023): 129. So similar are the manuscripts that the first sentences of the introduction from each are almost identical. While I acknowlege that the authors are pursuing a theme, they have somewhat undermined their own efforts.

In addition to the afore mentioned concerns, additional concerns are held for the experimental design. Specific comments are listed below.

Title: While this title matches the authors previous title, it is not entirely reflectitve of the associated data. Additionally 'galectinology' gives the impression that this manuscript gives an overview of galectins in placental disease, rather than reporting on specific galectin transcripts. Please consider reformating to include a more accurate title.

Amended.

Abstract:

-The abstract indicates three groups (ascending, norcardiaform and normal). However, the methods and results appear to compare normal and diseased regions of the same placenta.Please clarify throughout.

Clarified. 

- Please add P-values to the abstract for signficant findings.

Added.

Introduction:

- Please address concerns listed above with regard to self citation and heavy borrowing from your previous submission.

Amended

Methods:

Line 71: Extra zero in 450(0)

Amended.

Line 71: With such small numbers it is probably worthwhile to provide more information about the individual horses parity and age. Both of these play an important part in placental development and should not be brushed over.

While we understand the concerns for small numbers, the previous work published by El-Sheikh Ali and Murase indicated the superb clustering that these sample groups identified within.  We are not assessing placental development but rather response to placental disease within this manuscript.  Although we agree that age, parity, and even nutrition may play a role in the gene expression of the placenta, those variables are outside the confines of this study.  We leave to the editors discretion.

Lines 72-76: Please edit to improve readability. From abstract APL=7, NPL=4 and control=4. This is hard to follow in the methods as they are currently presented. This may also be improved by expanding your methods somewhat, rather than relying on the self citations.

Clarified.

Results:

- Please show PCA plots for all comparisons in both placentitis groups. Again it is unclear if the comparison of tissue is affected vs unaffected horse, or affected vs unaffected region. If it is the former then this study is grossly underpowered and the number of controls should be at least 2:1 to your cases to have robust comparisons. If it is the later then PCA by individual will help to demonstrate if disease truely drives difference in expression rather than the other variables (age, parity, sample location). 

We run PCA analysis for transcriptomic studies that are inclusive of thousands of transcripts to indicate how said transcripts cluster into different groups, but this is not appropriate for studies that investigate a limited number of targets.  The PCA analysis of this study has already been published in the cited studies (Murase 2021, Ali 2020).

-The  NPL gorup is really too small to perform statistical analysis and draw conclusions from. Consider including descriptive statistics only for this sextion.  Again please show a PCA even though it is limited by the sample size it is important to show.

We have amended the manuscript to include figures that should provide descriptive statistics.  Additionally, numerous peer-reviewed manuscripts have been published that utilized n=4 within the control group for this analysis. Additionally, we have added a paragraph to the discussion to allow readers to locate the PCA plots run on this dataset, as we feel it is inappropriate within the current manuscript.  We leave to the editors discretion.

- It would be more typical to show log2 fold changes for transcripts.

Amended.

Discussion:

Although interesting to read and well written, the discussion fails to address any limitations of the current study. The most glaring of these limitations is that this set of transcripts is repeated data from the original RNA-seq analysis (Ali et al Vet Rec 2021). The galectin transcipts for NPL have already been reported and are contained in the supplementary data of that publications. This current publications offers no new analysis of these data and simply repeats that which has already been reported. I acknowledge that this manuscript is highlighting galectins as a new avenue of study, however, more independent work or analysis should have been done (qPCR, protein work etc) to advance the field.

While we understand the reviewer’s comments, we disagree that this manuscript does not provide any additional information regarding galectin expression in placentitis.  All published transcriptomic studies must be accompanied by an excel document to file all DEGs within the study, to allow for broadening of our understanding of the gene expression within the horse.  While we are tasked with doing so, it does not limit us from investigating individual and select hypothesis within those datasets.  Additionally, the excel flle mentioned is not the same analysis, as it analyzed the data based on corrected false discovery rate (FDR), while this manuscript compared FPKM.  FDR will “lose” many DEF due to the correction methods, and FPKM allows us to increase sensitivity for these changes, which allowed for us to notice differences in galectin-8, -9, and -13- none of which came up as different in the excel sheet noted. 

We also add that due to read depth, and heightened sensitivity and specificity of RNAseq, PCR is no longer recommended to “validate” the findings.  Regarding protein work (IHC, western blots, etc) , we did not attempt to interpret any of our results as protein localization or production in this manuscript, but rather expression of genes.  We leave to the editor’s discretion.

The other major limitation is that of power. This study is very underpowered for the type of investigation. In particular there are far to few controls and this should be improved bfore resubmission.

Minimal variability was noted within controls, and a power analysis was performed prior to experimentation, showing that n=4 for controls was appropriate. We leave to editor’s discretion.

Reviewer 3 Report

Thank you for a well written manuscript and a thought provoking project.

There are just a few points that need to be re-worked. 

Line 55-56: you state that no etiology for focal mucoid placentitis has been confirmed. Do you mean pathogenesis? The etiology is nocardioform - actinomycetes such as Crossiela equi and others. 

In Table 1.  The legend states, "Red/green indicates an increased/decreased fold change in diseased placenta with red 123 indicating increase and green indicating decrease." However, the data on Galectin-8 has normal FPKM at 13.1 + 0.7 and the ascending placentitis at 5.6 + 0.5, a significant decrease , but the fold change of 2.3 is in red font.  Additionally Galectin-13 has normal FPKM at 1.7 + 0.5 and the ascending placentitis at 10.5 + 3.8, a significant decrease , but the fold change is listed as N/A. That appears, without the raw data, to be questionable. 

Line 153-154, listed alterations galectin -1, -3, -3BP, -9, -12, and -13, yet in Table 2. there is a decrease alteration in -8 of 1.8 fold (green font), and in Table 1. a fold change of 2.3, but that change may be a decrease, but it is in red font as a 2.3 fold increase.

Line 166-167, states that an increase in galectin-3, -9, and -13 was only found in mucoid placentitis, yet an increase from 1.7 + 0.5 to 10.5 + 3.8 was listed for galectin-13 in ascending placentitis in Table 1.  

Line 174, upregulation 'has' been found

Line 177-178, states with regards to galectin-13, a trend towards an increase was observed following mucoid placentitis, yet in Table 2. a significant 63.9 fold increase is listed, whereas in Table 1. ascending placentitis a N/A fold increase is listed next to a 13 1.7 + 0.5 to 10.5 + 3.8 increase. 

Line 185-186 and 206-207, galectin-8 decreased in both types, yet the font is red for ascending placentitis in Table 1. 

These confusing sentences need to be clarified. 

Author Response

Thank you for taking the time out of your busy schedules to review this manuscript.  We have responded to each individual comment, and our responses can be seen in bold.

Thank you for a well written manuscript and a thought provoking project.

There are just a few points that need to be re-worked. 

Line 55-56: you state that no etiology for focal mucoid placentitis has been confirmed. Do you mean pathogenesis? The etiology is nocardioform - actinomycetes such as Crossiela equi and others. 

We have not confirmed that etiology for nocardioform IS amycolatopsis or crossiella (Canisso et al.) – although we do culture/PCR those two bacteria from a subset of placentas every year, they have not been proven to cause the disease of focal mucoid placentitis. 

In Table 1.  The legend states, "Red/green indicates an increased/decreased fold change in diseased placenta with red 123 indicating increase and green indicating decrease." However, the data on Galectin-8 has normal FPKM at 13.1 + 0.7 and the ascending placentitis at 5.6 + 0.5, a significant decrease , but the fold change of 2.3 is in red font.  Additionally Galectin-13 has normal FPKM at 1.7 + 0.5 and the ascending placentitis at 10.5 + 3.8, a significant decrease , but the fold change is listed as N/A. That appears, without the raw data, to be questionable. 

N/A indicates that there was no significant difference, so fold change could not be assessed. The other comment was clarified/amended.

Line 153-154, listed alterations galectin -1, -3, -3BP, -9, -12, and -13, yet in Table 2. there is a decrease alteration in -8 of 1.8 fold (green font), and in Table 1. a fold change of 2.3, but that change may be a decrease, but it is in red font as a 2.3 fold increase.

Amended

Line 166-167, states that an increase in galectin-3, -9, and -13 was only found in mucoid placentitis, yet an increase from 1.7 + 0.5 to 10.5 + 3.8 was listed for galectin-13 in ascending placentitis in Table 1.  

Although an increase “appears” to have occurred for galectin-13 in ascending placentitis, this was not found to be significant, hence the N/A as fold change could not be assessed. A trend towards an increased was noted in focal mucoid placentitis (P=0.09) which has been clarified in  the abstract and the results. Additionally, the identifier ¥ is within the table and noted in the title/description.

Line 174, upregulation 'has' been found

Amended.

Line 177-178, states with regards to galectin-13, a trend towards an increase was observed following mucoid placentitis, yet in Table 2. a significant 63.9 fold increase is listed, whereas in Table 1. ascending placentitis a N/A fold increase is listed next to a 13 1.7 + 0.5 to 10.5 + 3.8 increase. 

Again, there was no significant difference noted in galectin-13 in ascending placentitis when statistics were performed. This is addressed above.

Line 185-186 and 206-207, galectin-8 decreased in both types, yet the font is red for ascending placentitis in Table 1. 

Amended.

These confusing sentences need to be clarified. 

Clarified.